# Biosynthesis of plant hemostatic dencichine in *Escherichia coli*

**Wenna Li[1], Zhao Zhou ®[1], Xianglai Li ®[1], Lin Ma[1], Qingyuan Guan[1], Guojun Zheng[1], Hao Liang[1], Yajun Yan ®[2], Xiaolin Shen ®[1], Jia Wang ®[1], Xinxiao Sun ®[1] ✉ & Qipeng Yuan ®[1] ✉**

Dencichine is a plant-derived nature product that has found various pharmacological applications. Currently, its natural biosynthetic pathway is still elusive, posing challenge to its heterologous biosynthesis. In this work, we design artificial pathways through retro-biosynthesis approaches and achieve de novo production of dencichine. First, biosynthesis of the two direct precursors $L-2$, 3-diaminopropionate and oxalyl-CoA is achieved by screening and integrating microbial enzymes. Second, the solubility of dencichine synthase, which is the last and only plant-derived pathway enzyme, is significantly improved by introducing 28 synonymous rare codons into the codon-optimized gene to slow down its translation rate. Last, the metabolic network is systematically engineered to direct the carbon flux to dencichine production, and the final titer reaches $1.29\,\mathrm{g\,L^{-1}}$ with a yield of $0.28\,\mathrm{g\,g^{-1}}$ glycerol. This work lays the foundation for sustainable production of dencichine and represents an example of how synthetic biology can be harnessed to generate unnatural pathways to produce a desired molecule.

Dencichine, or β-*N*-oxalyl-*L*-α,β-diaminopropionic acid (β-ODAP), is a plant active ingredient. It was first isolated from the seeds of *Lathyrus sativus*, and later was also found in *Panax notoginseng, Panax ginseng* and other plants[1]. β-ODAP can promote aggregation of platelets, and is the main hemostatic component of the famous Chinese medicine Yunnan Baiyao[2,3]. It is also effective in treatment of type II diabetic nephropathy, which is one of the most serious chronic complications of diabetes mellitus[4–6]. However, it is suspected to cause neurolathyrism that occurs upon prolonged over-ingestion of *L. sativus* seeds[7].

Currently, the supply of β-ODAP mainly relies on extraction from the root of *P. notoginseng*. However, its long growth cycle (3–5 years), requirement for special planting conditions (climate and soli), and low β-ODAP content limit the feasibility of this method for commercial application[3]. Chemical methods have been developed to synthesize β-ODAP from dimethyl oxalate and *L*−2, 3-diaminopropionate (*L*-DAP)[8], where *L*-DAP is prepared from the expensive carbobenzoxy-

*L*-asparagine. The use of concentrated acid/alkalis and toxic hydrogen sulfide makes the process environmentally unfriendly.

Biosynthesis has emerged as a promising alternative for sustainable production of chemicals, and an increasing number of plant secondary metabolites such as opioids[9], cannabinoids[10], breviscapine[11], icaritin[12], and scopolamine[13] have been successfully synthesized using metabolically engineered microorganisms. A major challenge to the heterologous production of β-ODAP is that its natural biosynthetic pathway has not been fully characterized yet. It was indicated that β-ODAP is formed by the condensation of *L*-DAP and oxalyl-CoA[14]. Only recently, the enzyme catalyzing this reaction (acyl transferase LsBAHD) has been identified by transcriptome sequencing and analysis of *L. sativus*[15]. However, the biosynthetic pathways for the two direct precursors (*L*-DAP and oxalyl-CoA) remain elusive in plants. Further elucidation of the natural pathways requires extensive high-throughput sequencing and multi-omics analysis. Alternatively, the continuous expansion of genetic databases and the in-depth understanding of

[1]State Key Laboratory of Chemical Resource Engineering, Beijing University of Chemical Technology, 100029 Beijing, China. [2]School of Chemical, Materials and Biomedical Engineering, College of Engineering, The University of Georgia, Athens, GA 30602, USA. ✉e-mail: sunxx@mail.buct.edu.cn; yuanqp@mail.buct.edu.cn

enzyme catalytic mechanisms enable the design and assembly of artificial pathways for a desired compound.

In plants, L-DAP is synthesized via a complex intermediate β-(isoxazolin-5-on-2-yl)-L-alanine (BIA)[16], but the exact route is still unclear. In microorganisms, L-DAP is a precursor of multiple secondary metabolites such as siderophore staphyloferrin B[17], antituberculosis drugs viomycin and capreomycin[18,19], and broad-spectrum antibiotic zwittermicin A[20]. In *Staphylococcus aureus*, L-DAP is synthesized from O-phospho-L-serine, and the reactions are sequentially catalyzed by SbnA and SbnB[17]. The other precursor oxalyl-CoA is an intermediate in oxalate degradation, and the conversion of oxalyl-CoA from oxalate is catalyzed by oxalyl-CoA synthetase (AAE)[21,22]. Oxalate is a common metabolite in plants and microorganisms with multiple physiological roles such as metal detoxification and deterrence to insect feeding[23]. In most plants, oxalate is formed via ascorbate degradation, but the relevant enzymes are unidentified yet[24]. By contrast, oxalate can be synthesized from glyoxylate catalyzed by glyoxylate dehydrogenase (Gloxdh) or from oxaloacetate by oxaloacetate hydrolase (Oah) in microorganisms[25].

Heterologous gene expression in *E. coli* often encounters the problem of low protein solubility, which could lead to waste of cell resources and formation of metabolic bottlenecks. LsBAHD is the only known dencichine synthase. We observed that the classic strategies including co-expression of molecular chaperones and fusion expression with solubilizing tags are ineffective in improving its solubility. In recent years, modulating the translation rate by selective introduction of synonymous rare codons (SRCs) to codon-optimized genes has become a promising strategy to facilitate functional protein folding[26–29]. Usually, special gene regions such as the first few codons[26] and the beginning of a β-strand[27] were selected for introduction of the SRCs.

In this study, by integrating the above genetic and enzymatic information, we design artificial biosynthetic pathways for β-ODAP and achieve its de novo production (Fig. 1a). The solubility issues are largely avoided by enzyme mining. To improve the solubility of LsBAHD, we simultaneously replace all the 28 isoleucine codons in the codon-optimized LsBAHD gene with the SRC AUA. Further, efficient biosynthesis of β-ODAP is achieved by deletion of the endogenous precursor-consuming pathways and synergy of two oxalate-producing pathways. This work represents an example of designing microbial cell factories for sustainable production of valuable compounds with unknown natural pathways.

## Results

### *De novo* biosynthesis of *L*-DAP

The β-ODAP pathway was divided into three modules: Module I, *L*-DAP biosynthesis; Module II, oxalyl-CoA biosynthesis; and Module III, β-ODAP biosynthesis (Fig. 1a). In *S. aureus*, *L*-DAP is a precursor of the siderophore staphyloferrin B. The genes related to staphyloferrin B biosynthesis form an operon *sbnABCDEFGHI*, among which *sbnA* and

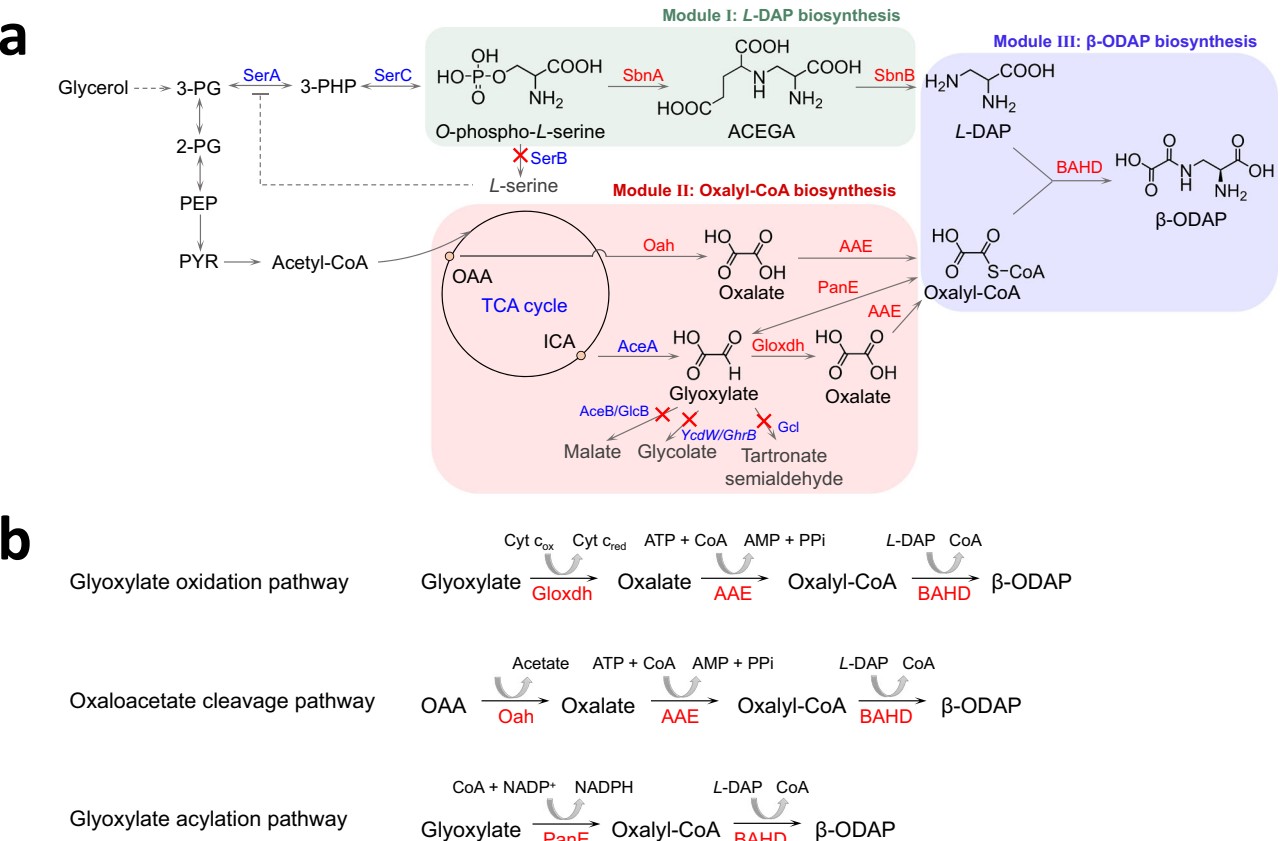

**Fig. 1 | Engineered biosynthetic pathways for synthesis of β-ODAP in *E. coli*.**
**a** Pathways for β-ODAP biosynthesis. **b** Three pathways for oxalyl-CoA biosynthesis in Module II. Endogenous enzymes are shown in blue and exogenous enzymes are shown in red. The blocked pathways are labeled with red crosses. Enzymes: AAE, acyl-activating enzyme 3; AceA, isocitrate lyase; AceB, malate synthase A; BAHD, dencichine synthase; Gcl, glyoxylate carboligase; GlcB, malate synthase G; Gloxdh, glyoxylate dehydrogenase; Oah, oxaloacetate hydrolase; PanE, oxalyl-CoA reductase; SbnA, *N*-(2 S)-2-amino-2-carboxyethyl-*L*-glutamate synthase; SbnB, *N*-(2 S)-2- amino-2-carboxyethyl-*L*-glutamate dehydrogenase; SerA, 3-phosphoglycerate dehydrogenase; SerB, phosphoserine phosphatase; SerC, phosphoserine aminotransferase; YcdW/GhrB, glyoxylate reductase. Metabolites: 2-PG, 2-phospho-D-glycerate; 3-PG, 3-phospho-D-glycerate; 3-PHP, 3-phosphooxypyruvate; ACEGA, *N*-(2 S)-2-amino-2-carboxyethyl-*L*-glutamate; ICA, isocitrate; *L*-DAP, *L*-2,3-diaminopropionate; OAA, oxaloacetate; PEP, phosphoenolpyruvate; PYR, pyruvate; β-ODAP, dencichine/β-N-oxalyl-*L*-α,β-diaminopropionic acid.

*sbnB* are responsible for *L*-DAP biosynthesis[17]. SbnA catalyzes the condensation of glutamate and *O*-phospho-*L*-serine to form *N*-(2*S*)−2-amino-2-carboxyethyl-*L*-glutamate (ACEGA) and SbnB is an NAD⁺-dependent dehydrogenase that hydrolyzes ACEGA to form 2-oxoglutarate and *L*-DAP. Strain BW1 (BW25113/pZE-sbnAB) over-expressing *sbnA* and *sbnB* accumulated 316.3 ± 12.6 mg L⁻¹ *L*-DAP in the M9Y medium at 48 h (Fig. 2a and Supplementary Fig. 1).

The phosphoserine phosphatase SerB is a key enzyme in *L*-serine biosynthesis and competes with SbnA for *O*-phospho-*L*-serine (Fig. 1a). Knockout of *serB* could reduce the shunt of *O*-phospho-*L*-serine and prevent the feedback inhibition of *L*-serine on *serA*. Indeed, compared with that of strain BW1, the *L*-DAP titer of strain BW2 (BWΔ*serB*/pZE-sbnAB) was increased by 166% to 842.5 ± 11.5 mg L⁻¹, although the cell growth was negatively affected and the cell density (OD₆₀₀) reached only 2.32 ± 0.03 at 48 h (Supplementary Fig. 1).

## Enzyme mining for Module II

After achieving *L*-DAP biosynthesis, we next started to establish efficient routes to oxalyl-CoA (Module II) (Fig. 1b). Oxalyl-CoA is an intermediate in oxalate degradation and is converted from oxalate by acyl-activating enzymes (AAEs). Foster[21,22] identified AAE from *Saccharomyces cerevisiae* (ScAAE) and *Arabidopsis thaliana* (AtAAE). The two enzymes show strict substrate specificity to oxalate, and ScAAE has better catalytic activity than AtAAE. In addition, there exists an acetyl-CoA: oxalate CoA-transferase YfdE in *E. coli*. However, the reaction is reversible and the enzyme has a high $K_m$ value for oxalate (22 mM)[30]. Therefore, ScAAE was selected for the synthesis of oxalyl-CoA from oxalate.

In microorganisms, two oxalate biosynthetic pathways have been reported, namely the glyoxylate oxidation pathway and the oxaloacetate cleavage pathway. In the former, glyoxylate is oxidized to oxalate by cytochrome *c* dependent glyoxylate dehydrogenase (Gloxdh) while in the latter oxaloacetate is cleaved into oxalate and acetate by oxaloacetate hydrolase (Oah). So far, only one Gloxdh has been characterized from the wood-rotting fungus *Fomitopsis palustris*[25]. We synthesized the codon-optimized gene *Fpgloxdh* and overexpressed it in *E. coli* BL21 Star (*DE3*). However, the protein existed mostly in the inclusion body (Supplementary Fig. 2a). To solve this problem, we searched the database for homologous proteins. A protein named Cyb2p from *S. cerevisiae* showed moderate sequence similarity with Fpgloxdh (50.3%). Cyb2p was previously identified as a *L*-lactate cytochrome *c* oxidoreductase that converts *L*-lactate to pyruvate[31]. To test its activity on glyoxylate, the 6×His-tagged Cyb2p was purified and subjected to in vitro assay. Interestingly, the result showed that Cyb2p exhibits aldehyde dehydrogenase activity besides alcohol dehydrogenase activity and catalyzes oxidation of glyoxylate to oxalate. Although the $V_{max}/K_m$ value of Cyb2p (0.15 × 10⁻⁴ s⁻¹ mg⁻¹ protein) is slightly lower than that of Fpgloxdh (0.17 × 10⁻⁴ s⁻¹ mg⁻¹ protein), the solubility of Cyb2p is much better than that of Fpgloxdh (Table 1 and Supplementary Fig. 2). Cyb2p was thus renamed as Scgloxdh and used for pathway assembly.

To obtain an efficient Oah, four candidates, namely Anoah from *Aspergillus niger*[32], Pcoah from *Penicillium chrysogenum*[33], Fpoah from *F. palustris*[25], Ssoah from *Sclerotinia sclerotiorum*[34] were selected and tested. The codon-optimized genes were successfully expressed in *E. coli*, and the corresponding proteins showed obvious differences in expression level and solubility (Supplementary Fig. 3a–d). Anoah and Ssoah existed mostly as inclusion bodies whereas Pcoah and Fpoah showed clear bands in the supernatants. As a result, only Pcoah and Fpoah were successfully purified and their catalytic activities were determined by in vitro assays. Compared with Pcoah, Fpoah showed lower $K_m$ value (0.42 versus 0.75 mM⁻¹) and similar $V_{max}/K_m$ value (9.05 versus 9.41 × 10⁻⁴ s⁻¹ mg⁻¹ protein; Table 1 and Supplementary Fig. 3e, f), and was thus selected for further pathway assembly.

Besides the above two pathways, we also used the glyoxylate acylation pathway to synthesize oxalyl-CoA. In *Methylobacterium extorquens* AM1, oxalyl-CoA reductase PanE is involved in the degradation of oxalate and catalyzes the reversible reaction between glyoxylate and oxalyl-CoA[35]. PanE was expressed and purified, and the results of in vitro assays showed that the $V_{max}$ and $K_m$ values of PanE are 14.98 × 10⁻⁴ mM s⁻¹ mg⁻¹ protein and 0.36 mM, respectively (Table 1 and Supplementary Fig. 4). Taken together, a total of three pathways for oxalyl-CoA synthesis were selected for further comparison (Fig. 1b).

## Activity analysis and solubility improvement of LsBAHD

Acylation is widely involved in the structural modification of natural products to improve their structural diversity, stability, and bioavailability[36–38]. There are two known acyltransferase families. The BAHD acyltransferases (BAHDs) use acyl-CoA thioesters as donor molecules whereas the Serine Carboxypeptidase-Like (SCPL) acyltransferases use 1-*O*-β-glucose esters instead[36]. LsBAHD was previously identified from *L. sativus* and its function has been confirmed by transient expression in *Nicotiana benthamiana*[15]. It contains the conserved HXXXD (residues 162–166) and DFGWG (residues 381–385) motifs of BAHD family. To the best of our knowledge, it is the only oxalyl-CoA transferase reported so far and has not been expressed in prokaryotic hosts. We performed phylogenetic analysis of BAHDs, and a total of 81 plant-derived proteins were selected, all having >50% sequence identity with LsBAHD. The result showed that the BAHDs are categorized into three groups (Clade I to III), and LsBAHD (QYL33117.1) has the closest relationship with that from *Medicago truncatula* (XP_013442394.1) with 64.35% sequence identity (Supplementary Fig. 5).

We synthesized the codon-optimized LsBAHD gene and overexpressed it in *E. coli* BL21 (*DE3*). The protein had poor solubility and was expressed mainly as the inclusion body (Supplementary Fig. 6a). The catalytic parameters of the purified enzyme were determined using a PanE-LsBAHD coupled assay, as the substrate oxalyl-CoA was not commercially available. The results showed that LsBAHD could effectively catalyze the condensation of oxalyl-CoA and *L*-DAP to form β-ODAP, and the $K_m$ and $V_{max}/K_m$ values for *L*-DAP are 0.52 mM and 0.92 × 10⁻⁴ s⁻¹ mg⁻¹ protein, respectively (Table 1 and Supplementary Fig. 6b, c). In addition, we also tested two other substrate combinations acetyl-CoA/*L*-DAP and oxalyl-CoA/*L*−2, 4-diaminobutyrate. LsBAHD was unable to catalyze these reactions, indicating that it has narrow substrate specificity.

After mining suitable enzymes, the oxalyl-CoA-forming pathways (Module II) and LsBAHD (Module III) were combined and their conversion capacities were evaluated by feeding experiments. Specifically, the genes required for module II and module III were cloned into plasmid pCS27 and transferred into strain BW25113 (BW, for short), generating strains BW3 (BW/pCS-LsBAHD-panE), BW4 (BW/pCS-LsBAHD-ScAAE-Scgloxdh) and BW5 (BW/pCS-LsBAHD-ScAAE-Fpoah), respectively. When fed with glyoxylate and *L*-DAP, BW3 produced non-detectable β-ODAP while BW4 produced 115.8 ± 17.3 mg L⁻¹ of β-ODAP at 48 h (Figs. 3a, b and 2b). The failure in β-ODAP production by strain BW3 may be attributed to the reversible reaction catalyzed by PanE, which is uncompetitive with the native enzymes for glyoxylate. When fed with oxaloacetate and *L*-DAP, BW5 produced 21.7 ± 3.3 mg L⁻¹ of β-ODAP at 48 h (Figs. 3c and 2b). Although β-ODAP was successfully produced, the titers were still low. Thus, we aimed to improve the production efficiency by removing the pathway bottlenecks.

First, LsBAHD was targeted as a rate-limiting step due to the poor solubility. To tackle this problem, we tried several conventional strategies such as co-expression with molecular chaperones (DnaKJ/GroSL/IbpAB) and fusion expression with soluble tags (maltose-binding protein/glutathione S-transferase). However, neither showed obvious effect (Supplementary Fig. 7). As aforementioned, the *LsBAHD* gene was codon-optimized to exclude *E. coli* rare codons. Codon

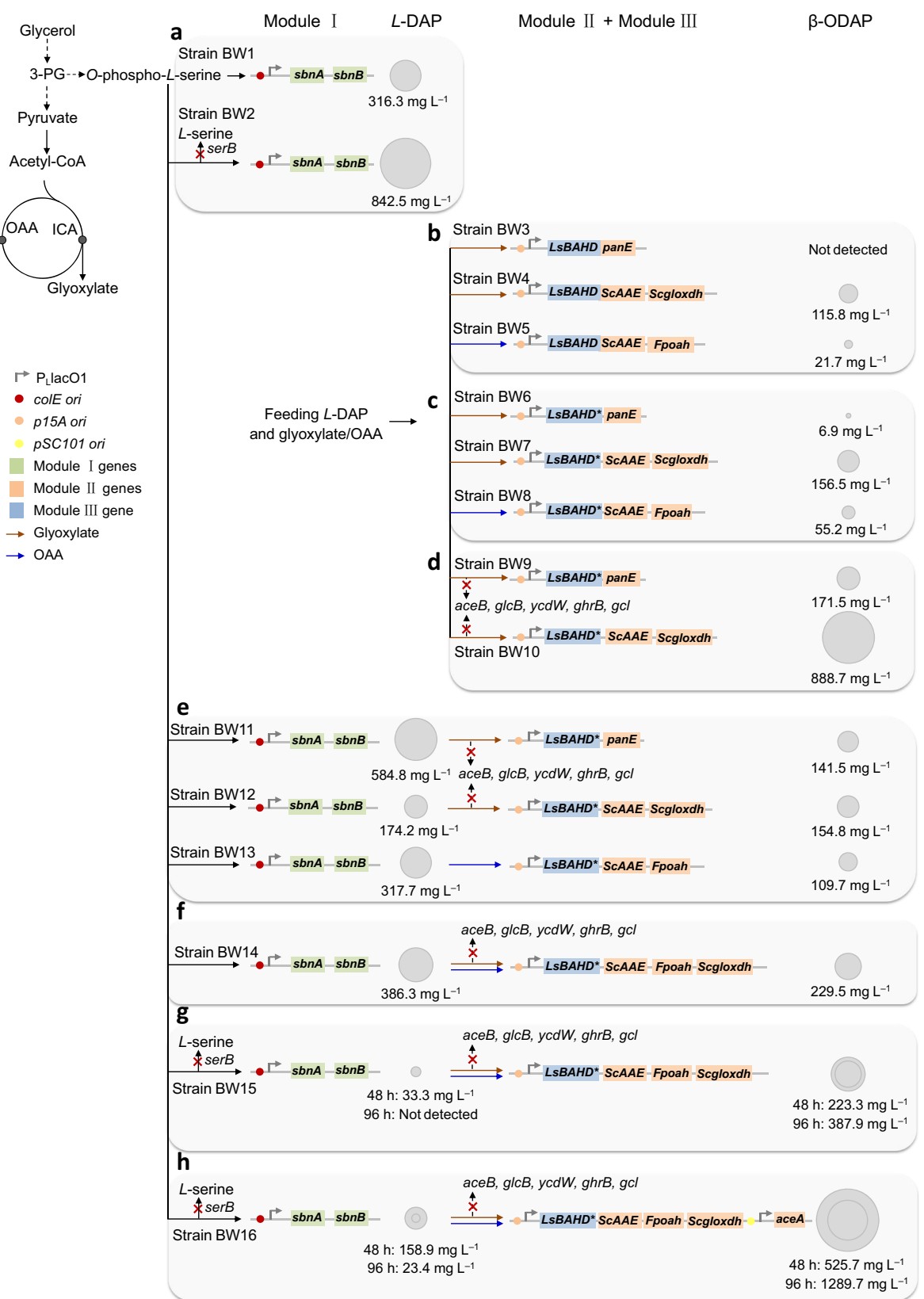

**Fig. 2 | Schematic of engineering strategies to optimize β-ODAP production in E. coli. a** Optimization of L-DAP production (Module I). **b** Comparison of the efficiency of different combinations in oxalyl-CoA (Module II) and β-ODAP (Module III) biosynthesis by feeding experiments. **c** Enhancing β-ODAP production by improving LsBAHD solubility. **d** Enhancing β-ODAP production by deleting glyoxylate degradation pathways (*aceB, glcB, ycdW, ghrB, gcl*). **e** De novo biosynthesis of β-ODAP by three different artificial pathways. **f** Enhancing β-ODAP production by synergy of the glyoxylate oxidation pathway and the oxaloacetate cleavage pathway. **g** Enhancing β-ODAP production by deleting *serB*. **h** Enhancing β-ODAP production by overexpressing *aceA*. The blocked pathways are labeled with red crosses. Solid arrows indicate single-step reactions, and dashed arrows indicate multi-step reactions. The area of the circles for L-DAP and β-ODAP is proportional to their titer.

**Table 1 | Kinetic parameters of exogenous enzymes in β-ODAP pathway**

| Enzyme | Organism | Substrate | $K_m$ (mM)[a] | $V_{max}$ (×10⁻⁴ mM s⁻¹ mg⁻¹ protein) | $V_{max}/K_m$ (×10⁻⁴ s⁻¹ mg⁻¹ protein) |
|---|---|---|---|---|---|
| Fpgloxdh | *Fomitopsis palustris* | Glyoxylate | 0.23 | 0.04 | 0.17 |
| Scgloxdh (Cyb2p) | *Saccharomyces. cerevisiae* | Glyoxylate | 0.74 | 0.11 | 0.15 |
| Pcoah | *Penicillium chrysogenum* | Oxaloacetate | 0.75 | 7.06 | 9.41 |
| Fpoah | *F. palustris* | Oxaloacetate | 0.42 | 3.80 | 9.05 |
| PanE | *Methylobacterium extorquens* AM1 | Glyoxylate | 0.36 | 14.98 | 41.61 |
| LsBAHD | *Lathyrus sativus* | L−2, 3-Diaminopropionate | 0.52 | 0.48 | 0.92 |

[a]The experiment was carried out in duplicate. The protein gels and fitting curves are shown in Supplementary Fig. 2, 3, 4, and 6.

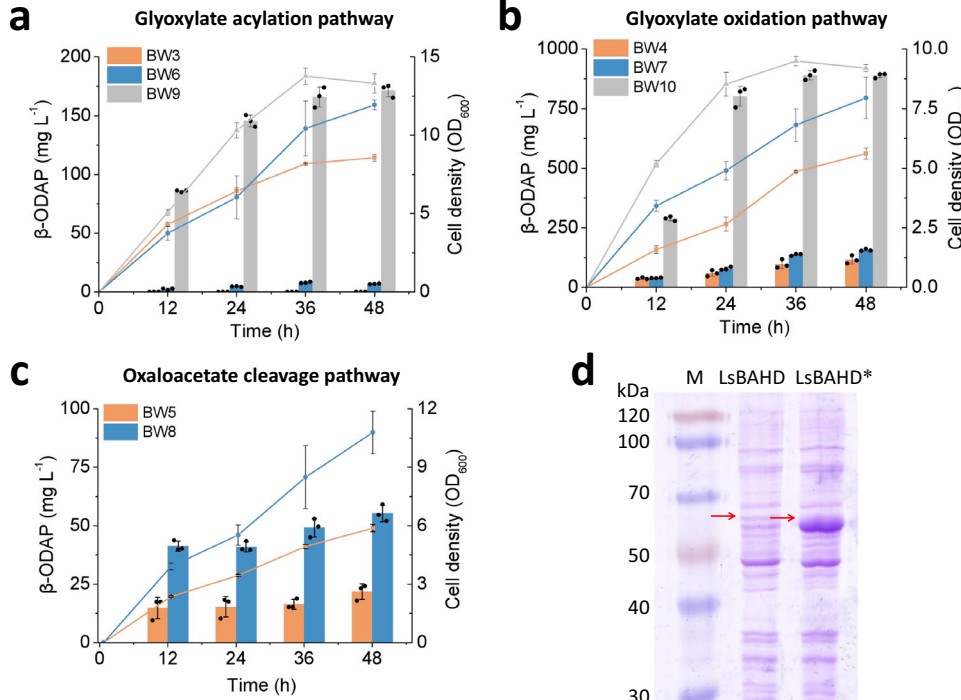

**Fig. 3 | Production and optimization of β-ODAP by feeding experiments.**
**a** glyoxylate acylation pathway; **b** glyoxylate oxidation pathway; **c** oxaloacetate cleavage pathway. The bars indicate the titer of β-ODAP and the lines indicate biomass at $OD_{600}$. Data shown are mean ± SD ($n = 3$ independent experiments).

**d** The solubility levels of LsBAHD and LsBAHD* were identified through SDS-PAGE analysis. Red arrows indicate the bands of LsBAHD and LsBAHD*. This experiment was repeated independently twice with similar results. Source data are provided as a Source Data file.

optimization has become a common strategy to improve heterologous gene expression. However, it may also cause protein misfolding and insolubilization due to the increase of translation rate and the elimination of proper translation pauses[39,40]. Studies have shown that selective introduction of SRCs could coordinate the co-translational folding of peptide chains, thereby enhancing functional protein expression[27,41,42]. In this study, we introduced SRCs of a certain amino acid throughout the protein instead of at certain locations. Isoleucine, which has the highest hydrophobicity among the 20 amino acids, was selected for codon replacement. Isoleucine is encoded by three synonymous codons AUU, AUC, and AUA, among which AUA is a rare codon in *E. coli*. The 28 isoleucine codons in LsBAHD-encoding gene were all replaced with AUA. As shown in Fig. 3d, the synonymous substituted gene (*LsBAHD**) resulted in approximately seven-fold increase in protein solubility compared with the original gene *LsBAHD*. This indicates that properly reducing protein translation rate by introducing SRCs is beneficial to promote protein folding and solubility. We replaced *LsBAHD* with *LsBAHD** in strain BW3, BW4 and BW5, generating strains BW6, BW7 and BW8. As a result, β-ODAP

production was significantly increased to 6.9 ± 0.3, 156.5 ± 4.8, and 55.2 ± 3.6 mg L⁻¹, respectively (Figs. 3 and 2c).

## Deleting glyoxylate degradation pathways enhances β-ODAP production

Glyoxylate is an intermediate in the glyoxylate cycle and can be metabolized by multiple native enzymes including malate synthases (encoded by *aceB* and *glcB*)[43], glyoxylate reductases (encoded by *ycdW* and *ghrB*)[44] and the glyoxylate carboligase (encoded by *gcl*)[45]. We stepwise knocked out the five genes in strain BW, leading to gradual decrease in the degradation rate without affecting the cell growth. The penta-knockout strain BWΔ5 completely consumed 1 g L⁻¹ of glyoxylate in 24 h while the wild type strain BW did in 8 h (Supplementary Fig. 8).

Plasmids pCS-LsBAHD*-panE and pCS-LsBAHD*-ScAAE-Scgloxdh were transferred into strain BWΔ5, generating strains BW9 and BW10, respectively. In the feeding experiments, the β-ODAP titers by the two strains reached 171.5 ± 5.4 mg L⁻¹ and 888.7 ± 9.9 mg L⁻¹, respectively, which are 24.88 and 5.68 times that of strain BW6 and BW7 (Figs. 3a, b

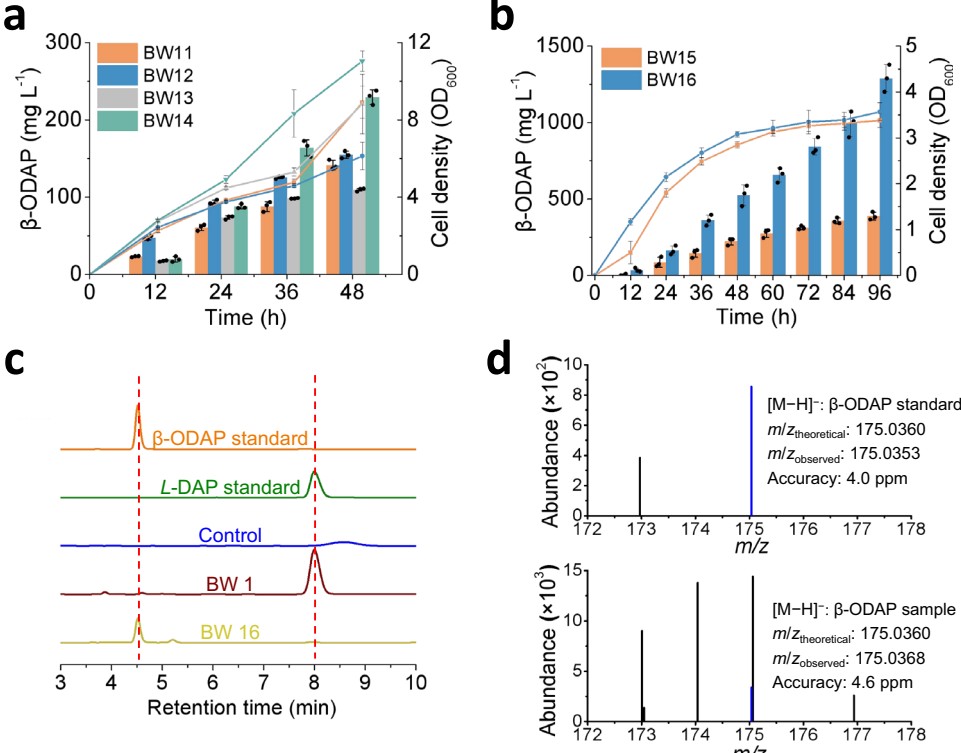

**Fig. 4 | *De novo* biosynthesis of β-ODAP. a** Comparison of β-ODAP production by three artificial pathways and synergy pathway. **b** Effect of knocking out of *serB* and overexpressing of *aceA* on β-ODAP production. The bars indicate the titer of β-ODAP and the lines indicate biomass at $OD_{600}$. **c** High-performance liquid chromatography (HPLC) analysis of the β-ODAP standard, *L*-DAP standard, the negative control strain (BW harboring empty vectors pZE12-luc and pCS27), the fermented product of the *L*-DAP-producing strain BW1 and the β-ODAP-producing strain BW16. **d** ESI-MS results of β-ODAP standard and sample (blue color). The $m/z_{theoretical}$ and $m/z_{observed}$ values noted are for the parent ions $[M − H]^{-}$. Data shown are mean ± SD ($n$ = 3 independent experiments). Source data are provided as a Source Data file.

and 2d). The results demonstrate that reducing glyoxylate degradation is beneficial to β-ODAP biosynthesis.

### Pathway synergy enhances de novo biosynthesis of β-ODAP

To explore de novo biosynthesis of β-ODAP, plasmid pZE-sbnAB was transferred into strain BW9, BW10, and BW8, generating strains BW11, BW12 and BW13, respectively. In shake flasks, the strains produced 141.5 ± 6.2, 154.8 ± 4.6, and 109.7 ± 1.9 mg L$^{-1}$ of β-ODAP with the accumulation of 584.8 ± 18.6, 174.2 ± 9.9, and 317.7 ± 4.1 mg L$^{-1}$ of *L*-DAP at 48 h, respectively (Figs. 4a and 2e). We further investigated the synergetic effect of the glyoxylate oxidation pathway and the oxaloacetate cleavage pathway on β-ODAP production. For this, strain BW14 (BWΔ5/pZE-sbnAB, pCS-LsBAHD*-ScAAE-Fpoah-Scgloxdh) was constructed, and as expected the β-ODAP titer was increased to 229.5 ± 9.3 mg L$^{-1}$ with the accumulation of 386.3 ± 23.7 mg L$^{-1}$ *L*-DAP. (Figs. 4a and 2f). This result demonstrates that oxalyl-CoA is the limiting precursor and pathway synergy is an effective strategy to increase its supply and β-ODAP production.

### Enhancing β-ODAP production by boosting the supply of glyoxylate and *L*-DAP

As shown above, inactivation of *serB* significantly improved *L*-DAP production. Accordingly, *serB* of BW14 was inactivated to construct BW15. Like the *L*-DAP producing strain BW2, the growth of BW15 was also negatively affected and the cell density ($OD_{600}$) reached only 2.85 ± 0.07 at 48 h. Compared with that of BW14, although the titer of β-ODAP by strain BW15 did not increase significantly at 48 h (223.3 ± 23.2 mg L$^{-1}$), it continued to increase and reached 387.9 ± 24.9 mg L$^{-1}$ at 96 h without *L*-DAP accumulation (Figs. 4b and 2g). As a result, the yield was increased from 0.02 to 0.12 g g$^{-1}$ glycerol.

Glyoxylate is a key precursor for the synthesis of oxalyl-CoA. In addition to reducing its degradation, increasing its supply may also be beneficial to β-ODAP biosynthesis. Isocitrate lyase encoded by *aceA* catalyzes the cleavage of isocitrate into succinate and glyoxylate. Gene *aceA* was overexpressed in strain BW15, generating strain BW16 (BW15 with pSA-aceA). The titer of β-ODAP produced by BW16 reached 1289.7 ± 88.8 mg L$^{-1}$ at 96 h with the accumulation of 23.4 ± 2.0 mg L$^{-1}$ *L*-DAP, which was 3.32 times that of BW15. The yield was further improved to 0.28 g g$^{-1}$ glycerol (Figs. 4b and 2h). The product of β-ODAP in the engineered strain BW16 was analyzed by HPLC and ESI-MS, and its retention time (Fig. 4c) and molecular weight (Fig. 4d and Supplementary Data 1) were in accordance with that of the β-ODAP standard The results demonstrated that boosting glyoxylate and *L*-DAP supply could increase the production of β-ODAP.

## Discussion

Biosynthesis has become a promising strategy for sustainable production of chemicals. A major obstacle to its broad application is the unavailability of the biosynthetic pathway for a target compound. So far, for numerous valuable natural metabolites, their biosynthetic pathways are still partially or completely uncharacterized. On one hand, many efforts have been devoted to elucidating the natural pathways with the help of multi-omics and bioinformatics analysis. On the other hand, the expansion of gene and enzyme databases enables the design of artificial pathways. This is usually achieved by establishing the cascaded transformation relationship between a target compound with an endogenous metabolite. In recent years, in silico approaches have been developed to accelerate and simplify this process[46,47]. In this

study, we achieved de novo biosynthesis of β-ODAP by designing artificial pathways. Noteworthy, we integrated the diversity in oxalate metabolism to direct more carbon flux to β-ODAP.

In a heterologous expression system, enzymes often encounter the problem of low solubility. For this, we adopted the strategies of enzyme mining and modulation of the translation rate by introducing SRCs. As indicated from Gloxdhs and Oahs, homologous enzymes from different species can exhibit dramatic different solubility. The underlying reason may be explained by sequence and structure comparison, which can provide useful guidance for further improvement. For LsBAHD, the SRC substituted gene resulted in significant increase in protein solubility and β-ODAP production. This whole-gene rare-codon substitution strategy may have potential application to improve solubility of other proteins.

## Methods

### Experimental materials

Strains and plasmids used in this study are listed in Supplementary Tables 1 and 2, respectively. Primers designed are listed in Supplementary Data 2. *E. coli* XL-1 Blue was used as the host for plasmid construction. *E. coli* BW25113 and its derived strains were used for feeding experiments and de novo production. *E. coli* strain BL21 Star (*DE3*) was used for protein expression and purification. Plasmids pZE12-luc, pCS27, and pSA74 were used for pathway construction. Plasmid pETDuet-1 was used as the vector for protein expression and purification. Plasmids were constructed by standard enzyme digestion and ligation. The knockout strains were constructed by λ-RED recombination following the standard protocols[48].

### Culture media and conditions

Lysogeny broth (LB) medium containing yeast extract ($5\,g\,L^{-1}$), tryptone ($10\,g\,L^{-1}$), and NaCl ($10\,g\,L^{-1}$) was used for inoculant preparation and cell propagation. Modified M9Y medium containing glycerol ($20\,g\,L^{-1}$), yeast extract ($2\,g\,L^{-1}$), $NH_4Cl$ ($4\,g\,L^{-1}$), $Na_2HPO_4$ ($6.78\,g\,L^{-1}$), $KH_2PO_4$ ($3\,g\,L^{-1}$), MOPS (morpholinepropanesulfonic acid, $2\,g\,L^{-1}$), NaCl ($0.5\,g\,L^{-1}$), $MgSO_4$ (1 mM), and $CaCl_2$ (0.1 mM) was used for shake flask cultivations.

For shake flask experiments, single fresh colonies were inoculated into 4 mL of LB media with appropriate antibiotics and grown overnight at 37 °C. Subsequently, overnight cultures (1 mL) were transferred to 250 mL shaking-flask containing 50 mL of fresh M9Y media, grown at 37 °C and 200 rpm for 2 h and then induced with 0.5 mM isopropyl-β-D-thiogalactoside (IPTG). The induced cultures continued to grow at 30 °C and 200 rpm. For the feeding experiments, *L*-DAP and glyoxylate/oxaloacetate were added together with IPTG, of which the strain BW10 was feed with $1\,g\,L^{-1}$, respectively, and the others were $0.5\,g\,L^{-1}$. Ampicillin, kanamycin, and chloramphenicol were added to the medium when necessary at final concentrations of 100, 50, and $34\,\mu g\,mL^{-1}$, respectively.

Samples were taken at regular time intervals for analysis of cell growth and product accumulation. The cell growth was monitored by measuring the optical density at 600 nm ($OD_{600}$) and the supernatants was subjected to HPLC analysis.

### In vitro enzyme assays

Plasmids pET-Fpgloxdh, pET-Scgloxdh, pET-Anoah, pET-Pcoah, pET-Fpoah, pET-Ssoah, pET-panE, pET-LsBAHD, and pET-LsBAHD* were transformed into *E. coli* BL21 Star (*DE3*), separately. The recombinant strains were cultured to an $OD_{600}$ of 0.6 and induced with 0.5 mM IPTG for 12 h at 25 °C. Cells were harvested and re-suspended in lysis buffer (50 mM Tris-HCl, 300 mM sodium chloride, 10 mM imidazole, pH 8.0). The His-tagged proteins were purified using $Ni^+$-affinity chromatography and protein concentration were determined using the bicinchoninic acid (BCA) method. The relative soluble protein levels of LsBAHD and LsBAHD* were quantified by grayscale scanning using the gel electropherogram analysis software Quantity One. $V_{max}$ and $K_m$ were determined by non-linear regression to the Michaelis–Menten equation.

The enzyme activity of Gloxdh was assayed with cytochrome *c* as a natural electron acceptor as follows. For Scgloxdh assay, the reaction system contains 2 mM cytochrome *c* and 0.062 μM of purified Scgloxdh in 100 mM $Na_2HPO_4$- $NaH_2PO_4$ buffer (pH = 7.5) with the final volume 0.5 mL. A gradient of glyoxylate concentrations (0.05–4 mM) was used to determine the $K_m$ and $V_{max}$. For Fpgloxdh assay, the reaction system contains 2 mM cytochrome *c* and 0.175 μM of purified Fpgloxdh in 100 mM $Na_2HPO_4$- $NaH_2PO_4$ buffer (pH = 7.5) with the final volume 0.5 mL. A gradient of glyoxylate concentrations (0.05–2 mM) was used to determine the $K_m$ and $V_{max}$. The activity of Scgloxdh and Fpgloxdh were determined by measuring the production of cytochrome *c* reduced at 550 nm from 0 to 100 s ($\varepsilon_{cytochrome\ c\ reduced} = 27.7 \times 10^3\,M^{-1}\,cm^{-1}$).

For Oah assay, the reaction system contains 0.18 mM $MnCl_2$ and 0.040 μM of purified Pcoah in 100 mM Tris-HCl buffer (pH = 8.0) with the final volume 0.5 mL. A gradient of oxaloacetate concentrations (0.05–2 mM) was used to determine the $K_m$ and $V_{max}$. For Fpoah assay, the reaction system contains 0.18 mM $MnCl_2$ and 0.053 μM of purified Fpoah in 100 mM Tris-HCl buffer (pH = 8.0) with the final volume 0.5 mL. A gradient of oxaloacetate concentrations (0.1–3 mM) was used to determine the $K_m$ and $V_{max}$. The activity of Pcoah and Fpoah were determined by measuring the consumption of oxaloacetate at 255 nm from 0 to 80 s ($\varepsilon_{keto-oxaloacetate} = 1.1 \times 10^3\,M^{-1}\,cm^{-1}$).

For PanE assay, the reaction system contains 0.5 mM CoA, 0.5 mM $NADP^+$ and 0.006 μM of purified PanE in 50 mM Tris-HCl buffer (pH = 7.5) with the final volume 0.5 mL. A gradient of glyoxylate concentrations (0.025–2 mM) was used to determine the $K_m$ and $V_{max}$. The activity of PanE was determined by measuring the production of NADPH at 340 nm from 0 to 80 s ($\varepsilon_{NADPH} = 6.22 \times 10^3\,M^{-1}\,cm^{-1}$).

Since the chemical oxalyl-CoA is not commercially available, a PanE-LsBAHD coupled assay was used to estimate the activity of LsBAHD. The reaction system contains 1 mM CoA, 1 mM $NADP^+$, 1 mM glyoxylate, and excess of purified PanE in 50 mM Tris-HCl buffer (pH = 7.5), which allow the glyoxylate converted to oxalyl-CoA in 30 °C. After 5 min, a gradient of *L*-DAP concentrations (0.125 - 1 mM) and 0.15 μM of purified LsBAHD were added to determine the $K_m$ and $V_{max}$ with the final volume 375 μL. The activity of LsBAHD was determined by measuring the production of NADPH at 340 nm from 0 to 200 s.

### HPLC analysis of product and intermediates

*L*-DAP and β-ODAP were analyzed by HPLC equipped with a reverse-phase Diamonsil C18 column (Diamonsil 5 μm, 250 × 4.6 mm) and UV−VIS detector. Cell culture samples were centrifuged at $7700 \times g$ for 10 min. The supernatant was reacted with 1-fluoro−2, 4-dinitrobenzene, filtered through 0.22 μm film and used for HPLC analysis. Solvent A was acetonitrile and solvent B was 0.5 M Acetic acid−Sodium Acatate buffer (pH = 4.5). The column temperature was set at 40 °C. The total flow rate was $1\,mL\,min^{-1}$, and the ratio of solvent A and solvent B was 17:83. Quantification was based on the peak areas at specific wavelengths (360 nm).

The analysis of glyoxylate and glycerol was performed by HPLC equipped with an Organic Acid Analysis Column (Amine HPX-87H Ion Exclusion Column, 300 mm × 7.8 mm) and refractive index detector. The mobile phase was 5 mM $H_2SO_4$ at a flow rate of $0.5\,mL\,min^{-1}$ and the oven temperature was set at 55 °C. The data of fermentation and enzyme assays were analyzed by the software OriginPro 9.0.

### Reporting summary

Further information on research design is available in the Nature Research Reporting Summary linked to this article.

 

## Data availability
Data supporting the findings of this work are available within the paper and its Supplementary Information files. A reporting summary for this Article is available as a Supplementary Information file. Source data are provided with this paper.

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

## Acknowledgements

This work was supported by National Key Research and Development Program of China (2021YFC2101500) and National Natural Science Foundation of China (21978015).

## Author contributions

XX.S. and Q.Y. designed the experiments. W.L. performed the experiments. Z.Z. and X.L. performed the enzyme assays. L.M. and Q.G. constructed plasmids. H.L., XL.S., and J.W. purified the protein. W.L. and XX.S. analyzed the data and wrote the manuscript. Q.Y., G.Z., and Y.Y. participated in the discussion and revision of the manuscript.

## Competing interests

The authors declare no competing interests.
