## [Peer Review File · Nature Communications]

Reviewers' Comments:

Reviewer #1:

Remarks to the Author:

This manuscript describes the generation of a synthetic pathway to enable *E. coli* to biosynthesize the plant metabolite dencichine. The authors propose a logical pathway whereby three modules are proposed: 1) biosynthesis of L-2,3-diaminopropionate (L-DAP), 2) biosynthesis of oxalyl-CoA, and 3) condensation of L-DAP and oxalyl-CoA to generate dencichine. For the production of L-DAP, the authors used the L-DAP biosynthetic enzymes from staphyloferrin B biosynthesis. This enables *E. coli* to produce large quantities of the amino acid, and in the process, they show that increased production can be accomplished by deleting a gene coding for an enzyme that competes for the needed phosphoserine precursor. They then focus on the second module, the formation of oxalyl-CoA. Here they analyze three different pathways from bacteria or yeasts as a source of this metabolite. They used solubility in *E. coli* and in vitro analyses of these enzymes to decide which to include in their studies. Since the final module has limited options, the authors focused on BAHD3 from the plant *L. sativus*. Initial studies analyzed BAHD3 which was produced using an *E. coli* codon-optimized version of the encoding gene. While this resulted in low levels of soluble protein, it allowed them to perform important preliminary analysis to determine whether the combination of modules II and III could produce dencichine from L-DAP and glyoxylate or oxaloacetate. They could also show improve production by inactivating genes in *E. coli* that coded for enzymes that would compete with their synthetic pathway for glyoxylate. Finally, all of this was put together to show they were able to generate a synthetic pathway that enables *E. coli* to generate dencichine at g/L levels from a simple growth medium.

This is an interesting study and the authors present a very logical progression in their thought process and pathway optimization. While this is a relatively simple pathway using mostly known enzymes, it is a solid example of how synthetic biology can be harnessed to generate unnatural pathways to produce a desired molecule. I have only minor comments.

1. In the abstract, the way it is worded, it gives the impression that the rare codon approach is something the authors developed. Clearly the authors know this was observed by others for improving the solubility of a protein of interest since they cite other work. It might be better to reword this to not give this impression.
2. Lines 93-94. It might be helpful to the reader if the authors note that the 28 Ile codons that were targeted are all of the Ile codons in the gene. I was wondering how the authors chose which ones to change and it wasn't until I looked up the protein that I realized the authors must have changed all the Ile codons. It would be helpful to not have readers go through that process.
3. Lines 176-177. I'm assuming the authors searched the databases for homologs proteins and not homologous genes. If so, I would suggest changing this sentence.
4. Lines 279-280. The wording of this sentence should be changed because I doubt the authors "expected" a seven-fold increase in soluble protein. They may have expected increased solubility, but not the exact fold increase.
5. Line 279-280. How did the authors quantify the increase in soluble protein levels? I may have overlooked this, but I do not think it was described. Also, the two arrows in 5b do not line up and appear to be pointing to different proteins.
6. Lines 295-296. The authors are discussing proteins but then list the gene names. This should be corrected.
7. Line 297. Change "theses" to "these"

8. Table 1. The authors need to present these data differently. First, they are reporting K_m values, but have not identified what substrate they are discussing for each enzyme. Second, there is an asterisk next to the K_m value for Fpgloxhdh, but no explanation of what that means.

9. Kinetic values. In the text and table, the authors report k_{cat} values and k_{cat}/K_m . In contrast, in the Materials and Methods, V_{max} is used not k_{cat} . This gets into an issue that is probably the most common error in the literature. The authors have used the BCA method to quantify their protein - I'm assuming they used BSA to make their standard curve. Due to this, it is incorrect to use k_{cat} values because the authors do not know the molar concentration of their protein. What they know is that their protein is at a mg/L concentration based on the BCA assay using an alternative protein as a standard. There is no evidence that their proteins behave in the BCA assay exactly like BSA. That is why the BCA and related assay provide you a mg/mL value, not a molarity value. Also, if you use a different protein as a standard, you can get a very different value. Therefore, the authors should only be reporting V_{max} and V_{max}/K_m values.

10. Kinetic values (part II). It would be helpful if the authors provided the standard error of the nonlinear regression analysis to obtain the kinetic parameters. It would also be helpful to understand how long the assays were performed and whether the assay conditions were determined to ensure they were run in the linear range of the assay.

11. Line 395. While LB is commonly thought to be an abbreviation for "Luria-Bertani" it is technically incorrect. As Bertani himself wrote, "For the historical record, the abbreviation LB was intended to stand for 'lysogeny broth.'"

12. Figure 2. While I found this figure to be quite informative, there are a couple of issues. First, the significant figures in the table are different than in the text. For example, BW2 in the figure makes 842.49 mg/L, but in the text the value is reported as 842.5 mg/L. Second, in sections e-g there is a reporting of the L-DAP values in these strains, but it is not noted in the text at all.

Reviewer #2:

Remarks to the Author:

This work designs artificial pathways through retro-biosynthesis approaches and achieves de novo production of dencichine for the first time. By optimizing two precursors, the expression of key enzyme BAHD3, pathway synergy, the final titer of dencichine reached 1.29 g L⁻¹. This study delivered some interesting results, but the organization of this MS is a mess. The figure 1 and 2 deliver too much information, but the figures 3, 4, 6, and 7 are too simple. It requires completely re-writing before considering for the journal.

Comments:

1. There is a suggestion that the introduction part could be more concise.
2. Please pay attention to the sentence tenses. Take several examples but not limited to them.
 - 1) Page 9 line 193-194: "The results of SDS-PAGE showed that all the four proteins are 194 expressed, but they have distinct expression levels and solubility."
 - 2) Page 11 line 240-242: "The result of SDS-PAGE showed that the protein can be expressed successfully, but mostly (81.43%) exists in the inclusion body, leading to a low purification yield."
3. The logic of the discussion section needs to be reorganized.
4. Page 17, line 418 "E. coli BL21 Star (DE3)" "DE3" is italic
5. All punctuations and marks should be in the same format.
6. Fig. 1b Please align the three lines. Fig. 4 is not clear enough.
7. BW15 is missed in figure 2.
8. BW11 produces more compounds than BW13, is there pathway synergy between them?

9. It is very weird that the yield of ODAP in BW9 is similar with BW11 when the module I was integrated. But under the same condition, the yield of ODAP decreased a lot from BW10 to BW12.
10. Why the overexpression of AceA in BW16 can rescue the default of cell growth in BW15?

Reviewer #3:

Remarks to the Author:

The manuscript by Li et al carried out successive manipulations and overexpression of native and foreign genes in *E. coli* to de novo synthesize a plant natural product, beta-ODAP. Authors systematically approached the yield improvement by engineering three segments, L-DAP, Oxalyl CoA, and beta-ODAP, separately. The engineering design and efforts described in this manuscript were exhaustive for all three components in the pathway, including disclosures of several negative data and some surprising data. It is unexpected to find that yeast Cyb2p enzyme can convert glyoxalate to oxalate in a relatively high efficiency. Also, replacing codon of all 28 Ile residues of BAHD3 to rare ones could significantly improve solubility, and thus overall productivity, was an impressive data. This is the first paper to de novo biosynthesize beta-ODAP in *E. coli* in a high yield (1.29 g per Liter) and new enzyme BAHD3 was discovered to complete pathway. Thus, this manuscript includes both scientific and engineering novelties. Several suggestions are described below to improve this manuscript.

Major comments

1. Qualitative data for beta-ODAP needs to meet the standard of analytical chemistry. Sup. Figure 7 showed ESI profile of standard and sample, but these are not sufficient to definitely prove that beta-ODAP is indeed biosynthesized in *E. coli*. Classical presentation is to provide LC chromatogram of selective ions for sample, negative control, and authentic standard with retention time. If High resolution-MS is used, delta ppm values of the sample in comparison to the standard needs to be given. Possibly MS/MS profiles of both sample and standard can be provided. Current presentation in Sup Fig 7 is not a proof of beta-ODAP production in *E. coli*.
2. In line 246, characterizations of the new enzyme BAHD3 were given in one sentence using a coupled assay. However, Michaelis-Menton kinetic properties cannot be deduced from the coupled assays for BAHD3 from the PanE and BAHD3 coupled assays as we do not know the exact oxalyl-CoA concentrations of BAHD3. In addition, LC-MS chromatograms for product formation from the coupled assays need to be provided with an appropriate negative control (e.g., boiled enzyme). Authors may simply present specific activity of the coupled assays together with LC-MS chromatograms (e.g., xx amount of beta-ODAP in specific reaction conditions).
3. In kinetic data in Sup Figure, standard deviations were calculated from two replicates. The minimum number of replicates for statistical analysis is three. Authors may simply present the mean values in the graph and state the data spread is less than xx% in duplicate experiments.
4. The approach for BAHD3 discovery in Fig 4 is confusing. Did authors know that *L. sativus* annotated gene models (or gene prediction from transcriptomic) were available in NCBI data base before they perform the similarity search? Which group sequenced and retrieved the transcript data from *L. sativus*? What type of Omics data (transcriptomics or whole genomics) from *L. sativus* have been available? Do you have a reference to add? More pre-existing knowledge context for BAHD3 needs to be given in the manuscript. Also, it is difficult to read texts in Figure 4. Author may place the Figure 4 in Sup Figure and find other ways to describe the description of BAHD3.
5. This is an extension of comment #3 above. BAHD constitute a large enzyme family even in a single species, and authors refers BAHD3 in the manuscript. Does that mean authors also find other BAHD homologues, such as BAHD1 and 2?

6. Line 274-275 – it is not clear why isoleucine is selected for SRC. Authors indicated isoleucine is the most hydrophobic residue, but why is the hydrophobicity the reason for SRC optimization?

Minor comments

There are dozens of grammatical mistakes. More thorough proofread is necessary.

L250: strong -> narrow

L270-271: add a reference to backup this sentence.

L281: benefit -> beneficial

L301: move "respectively" to the end of the sentence.

We greatly appreciate the review comments. We have carefully revised the manuscript and highlighted all the changes in yellow in the revised manuscript.

Reviewers' comments:

Reviewer #1 (Remarks to the Author):

This manuscript describes the generation of a synthetic pathway to enable *E. coli* to biosynthesize the plant metabolite dencichine. The authors propose a logical pathway whereby three modules are proposed: 1) biosynthesis of L-2,3-diaminopropionate (L-DAP), 2) biosynthesis of oxalyl-CoA, and 3) condensation of L-DAP and oxalyl-CoA to generate dencichine. For the production of L-DAP, the authors used the L-DAP biosynthetic enzymes from staphyloferrin B biosynthesis. This enables *E. coli* to produce large quantities of the amino acid, and in the process, they show that increased production can be accomplished by deleting a gene coding for an enzyme that competes for the needed phosphoserine precursor. They then focus on the second module, the formation of oxalyl-CoA. Here they analyze three different pathways from bacteria or yeasts as a source of this metabolite. They used solubility in *E. coli* and in vitro analyses of these enzymes to decide which to include in their studies.

Since the final module has limited options, the authors focused on BAHD3 from the plant *L. sativus*. Initial studies analyzed BAHD3 which was produced using an *E. coli* codon-optimized version of the encoding gene. While this resulted in low levels of soluble protein, it allowed them to perform important preliminary analysis to determine whether the combination of modules II and III could produce dencichine from L-DAP and glyoxylate or oxaloacetate. They could also show improve production by inactivating genes in *E. coli* that coded for enzymes that would compete with their synthetic pathway for glyoxylate. Finally, all of this was put together to show they were able to generate a synthetic pathway that enables *E. coli* to generate dencichine at g/L levels from a simple growth medium.

This is an interesting study and the authors present a very logical progression in their thought process and pathway optimization. While this is a relatively simple pathway using mostly known enzymes, it is a solid example of how synthetic biology can be harnessed to generate unnatural pathways to produce a desired molecule. I have only minor comments.

1. In the abstract, the way it is worded, it gives the impression that the rare codon approach is something the authors developed. Clearly the authors know this was observed by others for improving the solubility of a protein of interest since they cite other work. It might be better to reword this to not give this impression.

Response: Thanks for the comments. We deleted this sentence.

2. Lines 93-94. It might be helpful to the reader if the authors note that the 28 Ile codons that were targeted are all of the Ile codons in the gene. I was wondering how the authors

chose which ones to change and it wasn't until I looked up the protein that I realized the authors must have changed all the Ile codons. It would be helpful to not have readers go through that process.

Response: Thanks for the comments. The sentence is revised to make it more specific.

3. Lines 176-177. I'm assuming the authors searched the databases for homologs proteins and not homologous genes. If so, I would suggest changing this sentence.

Response: Thanks for the correction. The change is made as suggested.

4. Lines 279-280. The wording of this sentence should be changed because I doubt the authors "expected" a seven-fold increase in soluble protein. They may have expected increased solubility, but not the exact fold increase.

Response: Thanks for the comments. The sentence is revised as suggested.

5. Line 279-280. How did the authors quantify the increase in soluble protein levels? I may have overlooked this, but I do not think it was described. Also, the two arrows in 5b do not line up and appear to be pointing to different proteins.

Response: Thanks for the comment. The increase in soluble protein levels were estimated by grayscale scanning using the gel electropherogram analysis software Quantity One. The description is added in the Method section. Also, the two arrows in Fig. 5b are aligned.

6. Lines 295-296. The authors are discussing proteins but then list the gene names. This should be corrected.

Response: Thanks for the comments. We correct this in the text (Lines 279-281).

7. Line 297. Change "theses" to "these"

Response: Thanks for the correction. The change is made as suggested.

8. Table 1. The authors need to present these data differently. First, they are reporting Km values, but have not identified what substrate they are discussing for each enzyme. Second, there is an asterisk next to the Km value for Fpgloxhdh, but no explanation of what that means.

Response: Thanks for the comments. The substrate for each enzyme and the explanation of the asterisk are added in Table 1.

9. Kinetic values. In the text and table, the authors report kcat values and kcat/Km. In contrast, in the Materials and Methods, Vmax is used not kcat. This gets into an issue that is probably the most common error in the literature. The authors have used the BCA method to quantify their protein - I'm assuming they used BSA to make their standard curve. Due to this, it is incorrect to use kcat values because the authors do not

know the molar concentration of their protein. What they know is that their protein is at a mg/L concentration based on the BCA assay using an alternative protein as a standard. There is no evidence that their proteins behave in the BCA assay exactly like BSA. That is why the BCA and related assay provide you a mg/mL value, not a molarity value. Also, if you use a different protein as a standard, you can get a very different value. Therefore, the authors should only be reporting V_{max} and V_{max}/K_m values.

Response: Thanks for the comments. We revised the text and Table 1 to report the V_{max} and V_{max}/K_m values.

10. Kinetic values (part II). It would be helpful if the authors provided the standard error of the nonlinear regression analysis to obtain the kinetic parameters. It would also be helpful to understand how long the assays were performed and whether the assay conditions were determined to ensure they were run in the linear range of the assay.

Response: Thanks for the suggestion. The assays were performed in duplicate and thus the standard errors were not provided. Instead, the fitted curves were provided in the Supplementary information, and all the raw data points were shown in the curves. The reaction time for the enzyme assays was added in the Materials and methods section (100 s for Glox_{dh}, 80 s for Oah, 80 s for PanE and 200 s for BAHD). The reactions were all in the linear range at these timescales.

11. Line 395. While LB is commonly thought to be an abbreviation for “Luria-Bertani” it is technically incorrect. As Bertani himself wrote, “For the historical record, the abbreviation LB was intended to stand for ‘lysogeny broth.’”

Response: Thanks for the correction. “Luria-Bertani” is changed to “Lysogeny broth” in the text.

12. Figure 2. While I found this figure to be quite informative, there are a couple of issues. First, the significant figures in the table are different than in the text. For example, BW2 in the figure makes 842.49 mg/L, but in the text the value is reported as 842.5 mg/L. Second, in sections e-g there is a reporting of the L-DAP values in these strains, but it is not noted in the text at all.

Response: Thanks for the comments. We have rechecked the titers in Figure 2 to keep consistent with the main text. The amount of L-DAP accumulated by each strain was added in the text.

Reviewer #2 (Remarks to the Author):

This work designs artificial pathways through retro-biosynthesis approaches and achieves de novo production of dencichine for the first time. By optimizing two precursors, the expression of key enzyme BAHD3, pathway synergy, the final titer of dencichine reached 1.29 g L⁻¹. This study delivered some interesting results, but the organization of this MS is a mess. The figure 1 and 2 deliver too much information, but the figures 3, 4, 6, and 7 are too simple. It requires completely re-writing before considering for the journal.

Comments:

1. There is a suggestion that the introduction part could be more concise.

Response: Thanks for the suggestion. The introduction is revised to make it more concise.

2. Please pay attention to the sentence tenses. Take several examples but not limited to them. 1) Page 9 line 193-194: “The results of SDS-PAGE showed that all the four proteins are 194 expressed, but they have distinct expression levels and solubility.”

2) Page 11 line 240-242: “The result of SDS-PAGE showed that the protein can be expressed successfully, but mostly (81.43%) exists in the inclusion body, leading to a low purification yield.”

Response: Thanks for the suggestion. We checked the sentence tenses through the text.

3. The logic of the discussion section needs to be reorganized.

Response: Thanks for the suggestion. We completely reorganized the discussion section.

4. Page 17, line 418 “E. coli BL21 Star (DE3)” “DE3” is italic

Response: Thanks for the correction. The changes were made as suggested.

5. All punctuations and marks should be in the same format.

Response: Thanks for the suggestion. We checked this through the text.

6. Fig.1b Please align the three lines. Fig. 4 is not clear enough.

Response: Thanks for the comments. Fig.1b was revised to align the three lines. Fig. 4 was revised by removing the strain names.

7. BW15 is missed in figure 2.

Response: Thanks for the suggestion. BW15 is added in Fig. 2 as suggested.

8. BW11 produces more compounds than BW13, is there pathway synergy between them?

Response: Thanks for the comments. Strain BW11 and BW12 share the same precursor glyoxylate, while that of strain BW13 is oxaloacetate. To increase the supply of oxalyl-CoA, it is necessary to select pathways that consume different precursor when performing pathway synergy. It was evident that the glyoxylate oxidation pathway was more efficient than the glyoxylate acylation pathway in both feeding experiments (888.7 vs 171.5 mg L⁻¹) and de novo biosynthesis (154.8 vs 141.5 mg L⁻¹). Therefore, we chose the glyoxylate oxidation pathway and the oxaloacetate cleavage pathway for pathway synergy.

9. It is very weird that the yield of ODAP in BW9 is similar with BW11 when the module I was integrated. But under the same condition, the yield of ODAP decreased a lot from BW10 to BW12.

Response: Thanks for the comments. During the production process of the strain BW10, 1 g L⁻¹ of *L*-DAP and 1 g L⁻¹ of glyoxylate were added to the M9Y medium. But for the strain BW12, the introduction of plasmid pZE-sbnAB could supply *L*-DAP, there are no additional glyoxylate supplementation. Therefore, the production efficiency of strain BW12 may be limited by the insufficient supply of glyoxylate.

10. Why the overexpression of AceA in BW16 can rescue the default of cell growth in BW15?

Response: As shown in Fig. 6b, the cell growth of BW15 is similar with that of BW16.

Reviewer #3 (Remarks to the Author):

The manuscript by Li et al carried out successive manipulations and overexpression of native and foreign genes in *E. coli* to de novo synthesize a plant natural product, beta-ODAP. Authors systematically approached the yield improvement by engineering three segments, *L*-DAP, Oxalyl CoA, and beta-ODAP, separately. The engineering design and efforts described in this manuscript were exhaustive for all three components in the pathway, including disclosures of several negative data and some surprising data. It is unexpected to find that yeast Cyb2p enzyme can convert glyoxalate to oxalate in a relatively high efficiency. Also, replacing codon of all 28 Ile residues of BAHD3 to rare ones could significantly improve solubility, and thus overall productivity, was an impressive data. This is the first paper to de novo biosynthesize beta-ODAP in *E. coli* in a high yield (1.29 g per Liter) and new enzyme BADH3 was discovered to complete pathway. Thus, this manuscript includes both scientific and engineering novelties. Several suggestions are described below to improve this manuscript.

Major comments:

1. Qualitative data for beta-ODAP needs to meet the standard of analytical chemistry. Sup. Figure 7 showed ESI profile of standard and sample, but these are not sufficient to definitely prove that beta-ODAP is indeed biosynthesized in *E. coli*. Classical presentation is to provide LC chromatogram of selective ions for sample, negative control, and authentic standard with retention time. If High resolution-MS is used, delta ppm values of the sample in comparison to the standard needs to be given. Possibly MS/MS profiles of both sample and standard can be provided. Current presentation in Sup Fig 7 is not a proof of beta-ODAP production in *E. coli*.

Response: Thanks for the comments. The HPLC chromatograms of the sample, the negative control and the authentic standard were provided. And the delta ppm of the high resolution-MS was also calculated (Fig. 6c and 6d).

2. In line 246, characterizations of the new enzyme BAHD3 were given in one sentence using a coupled assay. However, Michaelis-Menton kinetic properties cannot be deduced from the coupled assays for BAHD3 from the PanE and BAHD3 coupled assays as we do not know the exact oxalyl-CoA concentrations of BAHD3. In addition, LC-MS chromatograms for product formation from the coupled assays need to be provided with an appropriate negative control (e.g., boiled enzyme). Authors may simply present specific activity of the coupled assays together with LC-MS chromatograms (e.g., xx amount of beta-ODAP in specific reaction conditions).

Response: Thanks for the comments. As suggested, the specific activity of the coupled assays was provided instead of the Michaelis-Menton kinetic parameters. The LC chromatograms were provided in Supplementary Figure 4b, as shown below.

3. In kinetic data in Sup Figure, standard deviations were calculated from two replicates. The minimum number of replicates for statistical analysis is three. Authors may simply present the mean values in the graph and state the data spread is less than xx% in duplicate experiments.

Response: Thanks for the comments. The error bars in the fitted curves are replaced with the raw data points. The data spread is also calculated and provided in the Sup Figures.

4. The approach for BAHD3 discovery in Fig 4 is confusing. Did authors know that *L. sativus* annotated gene models (or gene prediction from transcriptomic) were available in NCBI data base before they perform the similarity search? Which group sequenced and retrieved the transcript data from *L. sativus*? What type of Omics data (transcriptomics or whole genomics) from *L. sativus* have been available? Do you have a reference to add? More pre-existing knowledge context for BAHD3 needs to be given in the manuscript. Also, it is difficult to read texts in Figure 4. Author may place the Figure 4 in Sup Figure and find other ways to describe the description of BAHD3.

Response: Thanks for the comments. BAHD3 has been identified by transcriptome sequencing and analysis of *L. sativus* (Ref. 15). This was mentioned in the introduction section, and now is also referred in the Result section. Fig. 4 was revised by removing the strain names to make it clearer.

5. This is an extension of comment #3 above. BAHD constitute a large enzyme family even in a single species, and authors refers BAHD3 in the manuscript. Does that mean authors also find other BAHD homologues, such as BAHD1 and 2?

Response: Thanks for the comments. In the previous research (Ref. 15), 11 putative BAHDs (BAHD1 to BAHD11) were tested, and BAHD3 showed the desired activity. We now rename it to LsBAHD.

6. Line 274-275 – it is not clear why isoleucine is selected for SRC. Authors indicated isoleucine is the most hydrophobic residue, but why is the hydrophobicity the reason for SRC optimization?

Response: Thanks for the comments. A factor that leads to the formation of inclusion body is the interaction between the unfolded hydrophobic regions. Thus, we predicted that introducing SRCs of a hydrophobic amino acid such as isoleucine may be beneficial to the folding of the adjacent region.

Minor comments

There are dozens of grammatical mistakes. More thorough proofread is necessary.

L250: strong -> narrow

L270-271: add a reference to backup this sentence.

L281: benefit -> beneficial

L301: move "respectively" to the end of the sentence.

Thanks for the correction. The full manuscript was proofread to correct the grammar issues.

Reviewers' Comments:

Reviewer #1:

Remarks to the Author:

The authors have appropriately addressed the issues raised by the reviewers. The generation of a synthetic pathway in bacteria to generate a plant natural product will be of broad interest.

Reviewer #2:

Remarks to the Author:

The authors did not completely answer my questions.

1, "the figures 3, 4, 6, and 7 are too simple. It requires completely re-writing before considering for the journal. " The results in figure 3 are repeated with figure 2; There is not enough information in figure 4.

2."10. Why the overexpression of AceA in BW16 can rescue the default of cell growth in BW15?". I want to know the mechanism why AceA can rescue the phenotype.

Reviewer #3:

Remarks to the Author:

The revised manuscript addressed all critiques raised by this reviewer. The quality of the manuscript was improved.

Reviewers' comments:

Reviewer #1 (Remarks to the Author):

The authors have appropriately addressed the issues raised by the reviewers. The generation of a synthetic pathway in bacteria to generate a plant natural product will be of broad interest.

Response: Thanks for the reviewer's time and efforts in improving the quality of the manuscript.

Reviewer #2 (Remarks to the Author):

The authors did not completely answer my questions.

1, The results in figure 3 are repeated with figure 2; There is not enough information in figure 4.

Response: Thanks for the review comments. We moved Figure 2 and Figure 4 to the Supplementary information.

2. Why the overexpression of AceA in BW16 can rescue the default of cell growth in BW15. I want to know the mechanism why AceA can rescue the phenotype.

Response: Thanks for the reviewer's comments. As shown in Figure 4b, although strain BW16 grew better than strain BW15 at the early stage of cultivation.

According to the literature, activation of the glyoxylate cycle can redirect the isocitrate molecules directly to succinate and malate without CO₂ production, leading to the increased biomass yield (BMC Microbiol 2011, 11: 70). In this study, the slightly improved growth of strain BW16 by AceA overexpression may also be explained by the increased flux to the glyoxylate cycle.

Reviewer #3 (Remarks to the Author):

The revised manuscript addressed all critiques raised by this reviewer. The quality of the manuscript was improved.

Response: Thanks for the reviewer's time and efforts in improving the quality of the manuscript.

Reviewers' Comments:

Reviewer #2:

Remarks to the Author:

The authors have appropriately addressed the issues raised by the reviewers. The quality of the manuscript was improved.